# Application of Nanoparticles for Magnetic Hyperthermia for Cancer Treatment—The Current State of Knowledge

**DOI:** 10.3390/cancers16061156

**Published:** 2024-03-14

**Authors:** Marzena Szwed, Agnieszka Marczak

**Affiliations:** Department of Medical Biophysics, Institute of Biophysics, Faculty of Biology and Environmental Protection, University of Lodz, Pomorska 141/143 St, 90-236 Lodz, Poland; agnieszka.marczak@biol.uni.lodz.pl

**Keywords:** hyperthermia, magnetic nanoparticles, drug delivery systems, cancers

## Abstract

**Simple Summary:**

Hyperthermia (HT) is a commonly used technique applied as an effective sensitizer during cancer therapy. However, the localized heating of the tumor due to supraphysiological temperature may cause serious side effects towards normal tissues. Thus, new methods are needed to improve the precision of HT. Nanotechnology has allowed for the development of many promising tools to revolutionize traditional thermotherapy. Here, we showed that magnetic nanoparticles (MNPs) following activation by altered magnetic field not only destroy tumor cells but also cause increased blood flow to and oxygenation of cancer tissue. Moreover, we present the current state of knowledge regarding the combination of MNPs-based hyperthermia with traditional and innovative cancer therapies as well perspectives on its implementation within clinics.

**Abstract:**

Hyperthermia (HT) is an anti-cancer therapy commonly used with radio and chemotherapies based on applying heat (39–45 °C) to inhibit tumor growth. However, controlling heat towards tumors and not normal tissues is challenging. Therefore, nanoparticles (NPs) are used in HT to apply heat only to tumor tissues to induce DNA damage and the expression of heat shock proteins, which eventually result in apoptosis. The aim of this review article is to summarize recent advancements in HT with the use of magnetic NPs to locally increase temperature and promote cell death. In addition, the recent development of nanocarriers as NP-based drug delivery systems is discussed. Finally, the efficacy of HT combined with chemotherapy, radiotherapy, gene therapy, photothermal therapy, and immunotherapy is explored.

## 1. Introduction

Cancer is a generic term for various types of potentially malignant neoplasms resulting from genetic or epigenetic alterations to somatic cells. All cancers combined accounted for nearly 10 million deaths worldwide in 2020 [1]. Current cancer treatments include surgery, chemotherapy (CT), and radiotherapy (RT). However, these strategies are limited by systemic toxicity, the induction of multidrug resistance, and low efficacy. Moreover, highly toxic CT agents can cause hypoxia and lower pH, thereby promoting the proliferation of tumor cells [2]. Hence, novel and more effective anticancer agents are urgently needed to prolong disease remission and improve quality of life [3].

Among recently proposed non-invasive cancer therapies, hyperthermia (HT) has attracted considerable interest in oncology research [4]. Although the use of heat for cancer treatment dates back to 5000 BC, clinical studies were first reported in 1891 by William Coley who induced heat with extracts of *Streptococcus pyogenes*, later called “Coley’s toxin” [5]. Defined as the directional application of heat energy, HT is a rediscovered technique that can be combined with CT and RT. HT of solid tumors involves increasing the tissue temperature to 40–45 °C to initiate the coagulation of proteins and damage to other biological macromolecules in order to induce apoptosis. Also, HT can be used to trigger the activation of the immune response and improve the blood flow and oxygenation of tumor tissues, which have two-fold greater blood flow than normal tissues [6]. In contrast to HT, thermal ablation, which is typically performed at temperatures greater than 60 °C [7], is not suitable for maintaining the temperature of different areas of the tumor and can result in coagulative necrosis [8].

Even though a controlled method is lacking, HT is still a viable therapeutic option by incrementally increasing temperature with the use of nanoparticles (NPs). The physico-chemical properties of NPs are highly persistent in the tumor environment. However, most NPs used for the induction of HT are not biodegradable and, thus, cannot be removed from the system with other metabolic products [6]. Therefore, biodegradable NPs that can maintain their activities in biological fluids and various nano-delivery systems have been developed to prolong the half-life of transported materials, such as polymeric NPs for the distribution thermo-therapeutic nanosubstances within tumor tissues. HT can be induced by applying an altered magnetic field (AMF) with iron oxide NPs encapsulated in polymeric nanovesicles [9]. Locally generated HT facilitates the disintegration of nanocarriers, which have the ability to encapsulate a chemotherapeutic agent or radionuclide to eliminate cancer cells with minimal risk of damage to normal cells.

## 2. Types of Conventional HT Methods

Although HT has been applied for clinical cancer treatment, local control of temperature remains challenging. To date, three main types of HT have been applied in clinical practice (Figure 1): whole-body, regional, and local. Whole-body HT, which is often used as an adjuvant for metastatic disease, is applied with heating blankets and thermal chambers. Regional HT involves perfusion of the peritoneal cavity with heated fluids with anticancer drugs. However, the clinical application of both whole-body and regional HT is limited due to severe side effects, especially gastrointestinal symptoms (diarrhea, nausea, and vomiting) and cardiac complications (thrombosis, myocardial ischemia, and even myocardial failure) [10].

Local HT, a tumor-focused technique that can be applied at different stages of invasiveness, is classified into three types: external, luminal, and interstitial. External HT is characterized by surface-induced increases in temperature, so that the heat generated at the cutaneous layers reaches the superficial tumor. External heat generation can be achieved using microwaves, laser, radio-irradiation, or high-intensity ultrasound. However, the tumor and normal tissues are heated in a heterogeneous manner, as the temperature can range from 39 to 42 °C to lethal temperatures of 43–45 °C, which is a major disadvantage of this method. Luminal HT, which is primarily applied for the treatment of colorectal cancer, uses special probes placed as close as possible to the tumor located in the rectal lumen. Interstitial HT is based on heat generated via a metal antenna made of ferromagnetic material to increase the temperature of tumor tissues [11,12]. However, this method is highly invasive, painful, and can lead to necrosis as far as 1–2 cm from the applied heat source [13]. Since heat generated by conventional HT methods is distributed in a chaotic manner, improved methods are needed to minimize thermal damage to surrounding healthy tissues. Thus, novel HT methods based on magnetic NPs (MNPs) and carriers are discussed in the following sections.

## 3. Cellular and Molecular Aspects of HT

HT causes many changes within cells, resulting in the loss of cellular homeostasis and subsequent cell death [14,15]. The type and extent of cell damage is determined by temperature. Moderate temperatures of 39–42 °C are generally non-lethal, whereas temperatures > 42 °C can kill cells in a time-dependent manner [16]. Higher temperatures cause damage at the cellular level by facilitating unfolding of proteins, thereby exposing hydrophobic groups, resulting in protein aggregation (Figure 2). Proteins damaged by HT can also form aggregates with normal proteins. In addition, degradation of aggregated/misfolded proteins through the proteasomal and lysosomal pathways leads to compression within the nuclear matrix and irreversible changes to chromatin.

Heat-induced unfolding and aggregation of proteins also impact the nucleus, which contains large amounts of proteins and DNA. The direct cytotoxic effects of HT are due to the denaturation, aggregation, and degradation of specific proteins involved in DNA synthesis (DNA polymerases-α and -β), DNA repair, transcription, RNA processing, and translation, ultimately leading to cell cycle arrest and cell death [17,18,19]. Other effects of HT include disruption of the cytoskeleton, dysregulation of membrane permeability, and metabolic changes (e.g., uncoupling of oxidative phosphorylation) that lead to decreased energy production and increased intracellular levels of Na^+^, H^+^, and Ca^2+^ [20,21,22].

HT is a viable strategy for cancer treatment by either directly killing cancer cells or induction of sensitization to RT or CT. Although normal cells are more tolerant to heat, tumor cells exhibit stronger thermal cytotoxicity and can be selectively targeted due to the chaotic architecture of the vasculature and hypoxic and low pH regions of solid tumors [23]. Hence, HT is considered a noninvasive strategy that does not adversely affect normal cells [24,25]. HT can also trigger immune responses via several mechanisms. The goal of HT is to mimic a state of fever and activate the immune response to destroy cancer cells. At around 38.5 °C, the immune response is activated, while antitumor immunity is activated at 39–43 °C. Therefore, HT can be used in combination therapy and as an adjuvant immunotherapy [26,27,28]. Also, HT can indirectly modulate the innate and adaptive immune responses to target the tumor microenvironment by increasing the production of proinflammatory cytokines and heat shock proteins (HSPs), which activate antigen-presenting cells (APCs), such as dendritic cells (DCs) and macrophages [29,30,31].

HSPs have been implicated in the difference in sensitivity to HT between neoplastic and normal cells. HSPs are produced in response to stress, including heat [32]. Using special metallic culture vessels for the immediate and accurate regulation of temperature, Imashiro et al. [23] observed that, after increasing the temperature to 43 °C for 30 min, HSP72 expression was higher in normal human dermal fibroblasts than breast cancer (MCF-7) cells. In addition, HSP72 was localized in the nuclei of normal cells, which is associated with the development of thermotolerance, as compared to the cytoplasm of tumor cells, which results in apoptosis [33].

### 3.1. DNA Damage

HT can directly and indirectly damage DNA. As direct damage, HT induces breaks to single-stranded as well as double-stranded DNA, phosphorylates the C-terminal serine residues of histone H2AX and ataxia–telangiectasia-mutated protein, and downregulates the activities of DNA polymerases and topoisomerases. Indirectly, HT increases production of reactive oxygen species (ROS), arrests the cell cycle, and inhibits DNA replication, resulting in cell death. Moreover, HT promotes DNA damage in cancer stem cells, which can be beneficial against cancer because these cells are resistant to most classical treatment strategies [34]. In addition, HT causes irreversible DNA damage that can complement the effects of CT and RT [35]. Notably, exosomes extracted from heat-stressed tumor cells induce a bystander effect that can also cause DNA damage to tumor cells not exposed to heat stress [36]. Nonetheless, mapping of tumor-specific genetic aberrations by whole-exosome sequencing, in silico gene prediction, mass spectrometry, and T cell assays would be beneficial for identifying novel antigens [37].

### 3.2. HT-Induced Apoptosis

Apoptosis is a natural biological process which can control the proper development, homeostasis, and proliferation of new cells with constant removal of damaged and unnecessary cells. HT at 42–44 °C is reported to stimulate apoptosis of cervical cancer (HeLa) and leukemic cells [38]. However, some cell types are more tolerant to heat stress than others [39]. Major apoptotic pathways resulting in caspase activation include the extrinsic (death receptor mediated), intrinsic (mitochondrial mediated), and endoplasmic reticulum stress-mediated pathways (Figure 3). In the intrinsic signaling pathway, which is considered the primary pathway involved in HT the pro-apoptotic protein Bid (BH3 interacting-domain death agonist) regulates the translocation of the pro-apoptotic protein Bax (bcl-2-like protein 4) to the mitochondrial membrane. The ratio of endogenous pro- to anti-apoptotic proteins largely determines cell fate [40]. An elevated temperature alters the balance of pro- and anti-apoptosis Bcl-2 family proteins. Experiments performed with HeLa cells confirmed that HT at 42–43 °C led to the decreased expression of the anti-apoptotic proteins Bcl-2 and Bcl-xL (B-cell lymphoma-extra-large protein), and the increased expression of the pro-apoptotic proteins Bax, Bak (BCL2 antagonist/killer 1 protein), Puma (p53 upregulated modulator of apoptosis), and Noxa (phorbol-12-myristate-13-acetate-induced protein 1). HT also activates procaspase 9, the initiator of the intrinsic apoptosis pathway, and caspase 3, which promotes chromatin condensation. The significance of HT and the roles of Bcl-2 family proteins in apoptosis are demonstrated by reversal of the decreased translocation of Bax, Noxa, and Puma to the mitochondria and inhibition of cytochrome c release in cells tolerant to 40 °C. The Bcl-2/Bcl-xL inhibitor ABT-737 was reported to sensitize cells to apoptosis, indicating that Bcl-2 family proteins are involved in HT-induced apoptosis [40]. Also, HT in the range of 40 to 45 °C was shown to induce apoptosis of tumor cells via the intrinsic pathway, whereas higher temperatures resulted in necrosis [41]. During heat-induced cell death, caspase 2 forms a complex with the specific adaptor protein RIP-Associated Protein with the A Death Domain, which activates caspase 2 and cleaves Bid to tBid, causing changes to the mitochondrial outer membrane potential with the subsequent release of cytochrome c, resulting in the formation of apoptosomes consisting of cytochrome c, Apaf1 (apoptotic protease activating factor 1), and caspase 9 [25]. Moreover, HT activates the pro-apoptotic protein Bim (Bcl-2-interacting mediator of cell death), which induces apoptosis through a Bax/Bak-dependent pathway [42].

HT is also a strong activator of c-Jun N-terminal kinases (JNKs), which phosphorylate Bim to enhance pro-apoptotic activity [43]. HT-induced ROS production triggers apoptosis of various cell types [25]. HT can also trigger the extrinsic apoptosis pathways via activation of cell surface receptors (Figure 3). HT-induced apoptosis is partially dependent on activation of the Fas ligand, TNF-α (tumor necrosis factor α), and TRAIL (TNF-related apoptosis-inducing ligand) [39].

## 4. Use of MNPs with Locally Induced HT

Regardless of the type of HT, the use of different heat sources is accompanied by the generation of a specific temperature gradient within tumor tissues. Commonly used external heat sources, such as ultrasound and infrared radiation, have a narrow window of efficacy and could burn the skin surface before the temperature within the tumor reaches a satisfactory therapeutic level. Therefore, MNPs present an extremely promising tool for HT to penetrate the tumor while avoiding damage to healthy tissues [6,44]. The physicochemical properties of MNPs also support their use in HT. MNPs have a diameter of approximately 100 nm, with a large surface area to mass ratio, and are highly reactive. As an additional advantage, the irregular structure of MNPs facilitates the penetration of tumors [45].

The phenomenon of enhanced permeability and retention relies on the increased permeability of the blood and lymphatic vessels in the tumor environment. Thus, the large diameters of the intercellular spaces of tumors favor free localization, local accumulation, and prolonged retention of NPs. Localization near the vascularizing blood system of the tumor also favors the use of NPs for local HT. When a temperature gradient is created, maximum heat is induced at the vessel wall and decreases with distance from the perivascular space, which facilitates the destruction of the blood vessel network, reduces angiogenesis, and inhibits tumor metastasis. Since the main drawback of conventional HT is the lack of selectivity when heating the tissues, the application of NPs prevents the application of heat to healthy tissues situated along the path of external radiation [6]. NPs can absorb heat energy originating from an external source to enhance the effects of HT. In addition, NPs are the primary source of heat and reverse the direction of heat loss (inside-out HT). In this type of HT, NPs focus heat from an external source onto the tumor to induce localized thermal destruction, while minimizing damage to collateral tissues [3]. Thus, iron oxide NPs, gold NPs, and carbon nanotubes can be applied with inside-out HT (Figure 4).

For non-invasive HT, MNPs can localize heat generated with an AMF. As the most promising “thermo-sensitizer”, the use of iron oxide NPs in HT is supported by the size, ease of functionalization with both organic and inorganic compounds, biocompatibility, minimal toxicity, and ease of excretion [46]. In addition, the ferromagnetic properties of MNPs can be exploited for simultaneous therapeutic and diagnostic applications. Iron oxide NPs are characterized by theranostic action. MNPs are essential components of diagnostic systems used to monitor disease progression and responses to therapy. In addition, MNPs are used as components of complex drug delivery systems for anticancer drugs, immunomodulators, and nucleic acids [47].

When exposed to a magnetic field, the oscillatory vibrations of MNPs are accompanied by a local increase in temperature and the release of cargo, such as anticancer drugs [48,49].

In addition, MNPs are activated by an external magnetic field, through the magnetic coupling between their magnetic moment and the magnetic component of the field [50]. The energy from this coupling process is absorbed by MNPs and then released as heat. The heating capacity of MNPs follows from their magnetic properties [51]. It means that the heat generation is associated with dynamic hysteresis losses resulting from the magnetic moment relaxation of single domain nanoparticles. For this magnetic hyperthermia, biological tissues display no significant energy deposition, and this technique is safe for non-cancer cells [52].

Recent studies have focused on the magnetic dipoles of iron oxide-based NPs, such as superparamagnetic iron oxide nanoparticles (SPIONs). During AMF amplification, SPIONs align with the direction of the field and the resultant magnetic susceptibility is several folds greater than that of standard paramagnetic materials [53]. The synthesis of SPIONs usually involves precipitation of iron salts (mainly by photochemical methods) in the presence of ammonia, sodium nitrate, and sodium hydroxide [54]. Nonorganic SPION-type nanostructures are rapidly eliminated from the body. Therefore, various polymeric modifications are used to improve the biological performance of SPIONs, such as the combination of polyethylene glycol (PEG), β-cyclodextran, the non-ionic detergent Pluronic F127, and chitosan [55,56]. Surface modifications of SPIONs also involve the attachment of ligands, which act as vectors to increase the likelihood of direct delivery into the tumor niche. For example, folic acid–SPION conjugates can be attached to surface receptors for delivery of folate to tumors in the brain, breast, and liver [57]

The surface of SPIONs can also be modified with colloidal gold. Briefly, a thin layer of gold is heated by AMF to enhance the HT effect of the MNPs. The heat release is several folds greater with gold-coated SPIONs as compared to unmodified SPIONs. HT mediated by SPIONs can be combined with other cancer treatment strategies [58]. For example, SPIONs coated with thermosensitive polymers can be loaded with anti-cancer drugs and delivered into the tumor microenvironment. The application of an AMF increases the temperature within the tumor causing the destruction of SPIONs and the subsequent release of the drug [59].

The mode of intracellular transport influences the induced cytotoxic mechanisms of MNPs. Biocompatibility studies with human umbilical vein endothelial cells (HUVECs) showed that endocytosis is the main mechanism of the intracellular transport of dextran- and citric acid-coated SPIONs. Additionally, SPIONs were found to inhibit the migration and induce apoptosis of HUVECs [60]. An in vivo animal study revealed that the pharmacokinetics and biodistribution of MNPs were dependent on the mode of administration, hydrodynamic diameter, and surface charge. Iron oxide MNPs are preferentially accumulated in the liver and spleen, rather than the brain, heart, kidney, and lung. Moreover, the site of distribution is strongly correlated with the size of the MNPs, as large MNPs (up to 4 μm) are removed by the reticuloendothelial system (RES) of the liver, while those that are 200–250 nm are usually filtered by the spleen [44]. As the intravenous administration of MNPs accelerates their rapid elimination from the body, SPIONs should be loaded into cells ex vivo and infused directly within the tumor, as confirmed with neuronal progenitor cells loaded with SPIONs and transplanted into mice with induced melanomas. The progenitor cells penetrated the tumors and subsequent exposure to AMF resulted in significant tumor regression and prolonged survival [61]. Drug delivery systems are developed to maximize drug efficacy and minimize side effects of targeted therapies. Encapsulation of therapeutics or biologically incompatible particles in nanocarriers can increase the solubility and stability of delivered molecules, thereby increasing their half-life and bioavailability. Although significant progress has been made in the field of drug nanocarriers, widespread clinical use remains uncertain. Their interaction with living cells changes the surface reactivity of nanomaterials, which could lead to undesirable and unforeseen physiological consequences. Also, the physicochemical properties of drug delivery systems, such as size, shape, surface charge, and coating, determine the biocompatibility of nanotransporters [62]. The phenomenon of protein adsorption onto macromolecule surfaces, as described by Leo Vroman in 1962 [63], which is currently known as the “protein corona effect”, defines the biological identity of nanocarriers and may be important in the systemic biological response generated by NPs [64].

As the first nanocarriers approved by the U.S. Food and Drug Administration (FDA) for clinical use, liposomes are circular vesicles composed of a lipid bilayer surrounding an aqueous core [65]. Liposomal doxorubicin (DOX), marketed under the trade name Doxil^®^ in the USA and Caelyx^®^ in the European Union (Jonson and Jonson, New Brunswick, NJ, USA), was initially developed for the treatment of Kaposi sarcoma and multiple myeloma. As compared to non-encapsulated DOX, Doxil^®^ has greater efficacy and lower cardiotoxicity. However, the low stability of Doxil^®^ and the phenomenon of intravenous leakage, described as the uncontrolled release of DOX from liposomes into the bloodstream, significantly limited its widespread clinical use [66]. Hence, biodegradable polymeric NPs, which were developed as an alternative to liposomes, can improve therapeutic efficacy, while reducing the risk of adverse side effects. Nanocarriers based on polyelectrolytes are biocompatible and dissociate into polyanions and polycations in aqueous solutions. Biomacromolecules, such as nucleic acids, proteins, and polysaccharides, are classified as polyelectrolytes [67].

Various types of magnetic materials, including metal NPs, metal oxide NPs, and core–shell MNPs [68], are used for magnetic hyperthermia (MHT). The features of MNPs are strictly related to size, shape, composition, and structure, which are controlled by polymeric modifications during synthesis [49,69].

The polymer used as the coating material or as a component of the nanocarrier protects MNPs from oxidation and aggregation, while allowing for further functionalization in the body. However, even slight modification of the polymer could influence the adsorption of blood plasma proteins (opsonization) and their ability to bind to the membranes of macrophages and other cell types [70]. Opsonization usually leads to receptor-mediated phagocytosis and rapid clearance of MNPs from the blood, and facilitates the formation of a corona of plasma proteins, which increases the diameter of MNPs. For instance, MNPs with diameters larger than 200 nm are rapidly taken up by the RES and accumulate in the liver and spleen, while MNPs with diameters smaller than 6 nm are filtered by the kidneys [71]. Nevertheless, MNPs with diameters of 10–100 nm are pharmacokinetically ideal for in vivo applications [72,73]. MNPs can be encapsulated in a polymeric core shell or coated with suitable polyelectrolytes to form a protective layer against protein attachment. For example, iron oxide NPs coated with dimercaptosuccinic acid avoid opsonization and clearance by the RES, while reducing cell toxicity [70]. Moreover, surface functionalization with a hydrophilic PEG polymer can extend the half-life of iron oxide NPs to 12 h [74]. Polyvinylpyrrolidone and zwitterionic materials, such as dopamine sulfonate and poly(amino acids), as novel polymers, can provide a longer blood circulation time than PEG [75].

## 5. Bioconjugates of MNPs as Potential Passive or Targeted Delivery Systems

Modifications to and the size of the polymers within MNPs determine the mode of uptake by cancer cells. MNPs with diameters greater than 10 nm can extravasate and accumulate in tumor tissues, but not in normal tissues, due to natural differences between the defective and leaky vasculature of tumors and regular openings of normal vessels in healthy tissue [76]. The slower lymphatic clearance and venous return of the tumor microenvironment help to retain MNPs in tumor tissues [77]. The enhanced permeability and low clearance of solid tumors [68] can inhibit the targeted accumulation of MNPs [78], which is mostly due to the phagocytic activity of specialized cells of the RES [61]. In contrast to passive targeting, active targeting ensures accumulation of MNPs in solid tumor tissues via the overexpression of surface ligands specific to cancer cells [79]. Active targeting can facilitate the efficient internalization of targeted MNPs by receptor-mediated endocytosis [80] and has been described as a main factor affecting the binding of MNPs to cells in vitro [81]. Adhesion to the target cell is fully dependent on recognition of the targeting moiety of the MNPs [82]. Thus, the targeted internalization and accumulation of MNPs in cancer cells can avoid damage to normal cells. Antibodies, peptides, and ligands are the most commonly used molecules for targeting cancer cells.

The amount of MNPs delivered by active targeting could be insufficient to generate adequate heating at the tumor site [83]. However, many recent in vitro and in vivo studies [84,85] support the superiority of targeted versus non-targeted MHT. For example, the folate receptor (FR) facilitates the high-affinity binding of folate and related conjugates to target cells via receptor-mediated endocytosis [86]. A study by Li et al. [87] showed that the FR is required to mediate drug accumulation in targeted malignant cells.

Folic acid conjugates bind to the FR on the surface of malignant cells and form endosomes that are internalized to intracellular compartments [88]. Folic acid conjugation along with pH-sensitive linkers can increase the rate of drug secretion at pH 5.0 within growing cancer cells [89]. However, a linker is not necessary, as described by Bonvin et al. [90], who developed a folic acid-based theragnostic platform for simultaneous diagnosis and treatment of metastatic prostate cancer by targeting the lymph nodes [90]. Folate-conjugated SPIONs were reportedly taken up by FR-positive HeLa cells and significantly decreased the intensity of non-specific signals, thereby improving the accuracy of magnetic resonance imaging (MRI) [91]. While the FR is overexpressed by various cancer cells and minimally expressed by normal cells [92], the receptor tyrosine–protein kinase erbB-2 (HER2) is specifically expressed in some aggressive types of breast and lung cancers [93]. MNPs are used as initial nanocarriers and modified via PEGylation followed by the immobilization of trastuzumab (TRA), a monoclonal antibody against HER2, to target various types of breast cancer cells. Recently, Hamzehalipour et al. [94] described a novel drug delivery system (MNP-PEG-TRA) for the targeting of SK-BR-3 breast cancer cells in a mouse model of 7,12–dimethylbenz(a)anthracene (DMBA)-induced breast cancer. As compared to non-labelled MNPs, the proposed MNP-PEG-AMF drug delivery system remarkably enriched the effect of HT in cultured SK-BR-3 cells in vitro as well as DMBA tumor-bearing mice in vivo. The dosage of MNP-PEG-TRA was four-fold greater at the tumor site as compared to other organs, confirming considerable potential for treatment of breast cancer [94]. A cell-targeting function could also be added to thermally responsive core–shell MNPs by grafting a monoclonal antibody to target overexpressed HER receptors on mouse bladder tumor cells. In fact, the viability of mouse bladder tumor cells treated with the combination of fluorouracil, a cell-targeting agent, and an AMF was reduced by 50%. Interestingly, the same treatment, but with a non-specific targeting ligand (immunoglobulin G) or free fluorouracil, did not reduce cell viability as significantly [95]. Epidermal growth factor receptor (EGFR) is another surface protein that mediates cellular responses to various growth factors and is highly expressed on the surface of non-small cell lung cancer cells in almost 80% of patients [96]. EGFR is a useful diagnostic and therapeutic biomarker. In addition, EGFR-targeted SPIONs exhibited increased retention in tumor cells and moderately suppressed the growth of lung tumors [97].

Since membrane proteins, such as EGFR, can be targeted with antibodies and other molecules, it should be possible to entice the cell to internalize the antibody–antigen–MNP complex [98]. After the application of an AMF, MNPs are endocytosed and may interact with different membrane compartments, which (1) internalize MNPs from the plasma membrane, (2) recycle the MNPs back to the surface via early and recycled endosomes, or (3) degrade the MNPs via late endosomes and lysosomes. Increasing subcellular temperatures may induce cell death through lysosomal death pathways, suggesting potential applications of MNPs to induce death of apoptosis-resistant cancer cells. Clerc et al. [99] reported that the induction of magnetic intralysosomal hyperthermia caused cell death through a non-apoptotic signaling pathway by locally increasing temperature. Specifically, gastrin-grafted MNPs delivered to lysosomes induced generation of free radicals via the Fenton reaction. Subsequently, magnetic intralysosomal hyperthermia triggered permeabilization of the lysosomal membrane, resulting in the leakage of lysosomal enzymes into the cytosol, including cathepsin B, which activated non-apoptotic caspase 1. In contrast, Domenech et al. [100] reported that the application of an AMF induced the accumulation of EGFR-targeted MNPs resulting in the subsequent disruption of lysosomes, but no local increase in temperature, although ROS production was increased and cell viability was reduced. Shah et al. [101] developed a magnetic core–shell NP to deliver a mitochondria-targeting pro-apoptotic amphipathic tail-anchoring peptide to cancer cells combined with increased localized temperatures, which significantly enhanced apoptosis due to a synergistic effect on mitochondrial dysfunction in cancer cells [101].

## 6. Combination Therapies Using NP-Based MHT

One of the most clinically significant factors concerning the application of MHT is the effect of increased temperature on blood flow within the tumor and the surrounding healthy tissues [51]. The blood vessels are among the first barriers to interact with MNPs. Thus, increased permeability of vascular walls or blood vessel dilation is a natural consequence of constant exposure of non-cancer endothelial cells to elevated temperatures (41–46 °C). This issue can be addressed by the synergistic application of MHT together with CT, RT, photothermal therapy, or gene therapy, which are also more effective when combined with other modalities [9].

### 6.1. CT

MHT in conjunction with CT provides several advantages, including increased intracellular drug concentrations, inhibition of DNA repair, and a reduction in the proportion of apoptosis-resistant cancer cells [102]. Local application of MNPs with MHT can increase blood flow to effectively accelerate intracellular drug delivery and release in cancer tissues. In this scenario, the initial application of MHT will augment the CT-triggered death rate of already dying cancer cells. Ideal polymers for development of MNPs should be controlled by pH, temperature, or AMF to release loaded therapeutics [103]. Polymeric MNPs can be loaded with various classes of anti-cancer drugs, including hydrophilic agents, such as the anthracycline DOX, which is water-soluble at a mildly acidic or neutral pH [104]. The major molecular mechanisms underlying the anticancer activities of DOX include intercalation into DNA, p53-induced apoptosis, ROS production, and mitochondrial dysregulation. However, under continuous oxidative stress, DOX can also act on normal, noncancerous tissues, thereby impairing the function of healthy organs, especially the heart, liver, and kidney [105]. Although approved by the FDA for cancer therapy, the liposomal nanoformulation Doxil^®^ did not significantly improve the quality of life of cancer patients [106]. Mai et al. [107] engineered magnetic thermoresponsive iron oxide nanocubes (TR-cubes) for use with MHT for heat-mediated drug delivery. Iron oxide-based NPs were selected due to their outstanding stability and performance with MHT. Copper-mediated polymerization with ultraviolet light increased the polymerization rate and prevented aggregation of the TR-cubes. Moreover, the TR-cubes were sufficient for the delivery of DOX, while maintaining thermo-responsiveness. The results of an in vivo study showed that DOX-loaded TR-cubes achieved complete tumor regression and the highest survival rate of animals exposed to an AMF. Polymeric MNP-based complexes can not only be applied for DOX delivery systems, but also possess additional features as promising tools for imaging. Thirunavukkarasu et al. [108] developed magnetic field-inducible drug-eluting nanoparticles (MIDENs) by encapsulating superparamagnetic iron oxide NPs and DOX in a temperature-responsive poly(lactic-co-glycolic acid) (PLGA) nanomatrix that, when exposed to an external AMF, generated heat at >42 °C, which subsequently triggered the controlled release of DOX from the nanomatrix. The results of an in vitro study showed that MIDENs exposed to an AMF effectively killed CT26 colon cancer cells. Moreover, the results of an in vivo T2-weighted MRI study indicated that the use of MIDENs exposed to an AMF suppressed the growth of malignant tumors [108].

Hybrid nanogels are also attractive nanocarriers for biomedical applications. These nanostructures are composed of thermoresponsive polymers and superparamagnetic NPs that can take up and release large amounts of DOX. Cazares-Cortes et al. [109] developed biocompatible, pH-responsive, magnetoresponsive, and thermoresponsive nanogels (MagNanoGels), composed of polymer complexes of maghemite (γ-Fe_2_O_3_) MNPs loaded with DOX. In PC3 prostate cancer cells, not only did the DOX-MagNanoGels efficiently internalize DOX, but also intracellular release of DOX could be remotely triggered by an AMF, thus improving the cytotoxic properties of DOX [109].

Since DOX is a key chemotherapeutic agent, different types of porous magnetite nanospheres (PMNS) conjugated with DOX have been evaluated for the treatment of early and advanced breast cancer. For instance, Sharifi et al. [110] reported that lactoferrin-DOX-PMNS significantly suppressed the proliferation of 4T1 breast cancer cells and reduced tumor weight by prolonging drug availability and potential drug loading in cancer cells. Other antineoplastic agents for breast cancer therapy can also be effectively encapsulated in magnetic-based drug delivery systems [111]. Polymeric micelles are multifunctional MNPs that allow for control of drug release via an external AMF and prevent early clearance by the RES. Zheng et al. [112] synthesized hyaluronic acid-C16 copolymers via a peptide formation process with subsequent co-encapsulation of the therapeutic agent docetaxel and SPIONs to form multifunctional micelles with specific targeting capability based on CD44 receptor-mediated endocytosis that was enhanced in the presence of an external magnetic field. The cytotoxicity of the anti-breast cancer drug mitoxantrone was improved by linking it to magnetic thermoresponsive copolymer NPs. Li et al. [113] reported the superb paramagnetic behavior of magnetic thermoresponsive copolymer NPs for controlled drug release, which improved the anti-tumor efficacy of mitoxantrone with fewer side effects.

Polymeric MNPs have been applied for successful targeted delivery of various anticancer drugs and many natural substances that possess antineoplastic properties for potential use with MHT. For instance, curcumin (CUR) has been shown to effectively inhibit tumor growth and has been classified as “Generally Recognized as Safe” by the FDA [114]. Senturk et al. [115] developed multi-functionalized NPs composed of SPIONs coated with a PLGA-PEG di-block copolymer and loaded with CUR that exhibited high antitumor activity against glioblastoma cells and confirmed that these MNPs generated heat at ~42–45 °C by exposure to an AMF for 15 min.

Novel polymeric magnetic nanosystems have been developed to deliver CUR alone and in combination with specific inhibitors. Sudame et al. [116] successfully encapsulated CUR and the calcium channel blocker nifedipine within d-ɑ-tocopheryl-based polymeric magnetic NPs to improve the magnetothermal effects and effectively release drugs in HepG2 cancer cells, which significantly reduced tumor growth [116].

### 6.2. RT

RT can damage the DNA of tumor cells but does not reduce the risk of metastasis [117]. MNPs can act as both HT agents and radiosensitizers [3,118]. Rezaie et al. [119] found that the combination of RT (2 Gy at 6 MV) and HT (43 °C for 1 h) with iododeoxyuridine-loaded PCL-PEG-coated MNPs significantly reduced colony formation of U87MG glioblastoma spheroids in culture. Grauer et al. [57] assessed the efficacy of SPIONs combined with HT (six 1-h courses of SPION-mediated HT) and stereotactic RT (39.6 Gy) for treatment of recurrent glioblastoma multiforme and found that intracavitary HT in combination with RT induced a powerful antitumor immune response, mostly around the resection cavity, and achieved long-term stabilization [57]. The combination of MNP-HT and RT has also been successfully applied for different types of prostate cancer. For example, Attaluri et al. [120] reported that RT at 5 Gy in combination with MHT with polymeric bionised nanoferrite NPs for 1 h improved efficacy against both LAPC-4 and PC3 prostate cancer cells as compared to either RT or HT alone. Also, outstanding results were achieved against PC3 and LAPC-4 cells using mouse models of human prostate cancer. Another animal study found that HT (43 °C for 20 min) plus RT (5 Gy) extended survival by two-fold as compared to the control group. Moreover, Jiang et al. [121] developed gadolinium-doped iron oxide NPs with improved superparamagnetic properties and higher specific absorption rates than ordinary iron oxide, which successfully delayed the progression of prostate C1 tumor growth by 10 days in a transgenic mouse model of adenocarcinoma treated with MHT and RT as compared to mice treated with only MHT (2.5 days) or RT (4.5 days). Moreover, immunohistochemical staining revealed diminished hypoxia with vascular disruption, suggesting reduced resistance to radiation [121].

### 6.3. Gene Therapy

Polymeric MNPs are promising tools for the targeted delivery of foreign genetic material (oligonucleotides, genes, or gene segments). The many advantages of MNPs include improved solubility, pharmacology, and stability [122]. In addition, gene delivery with MNPs can restore normal cellular functions and mediate programmed cell death [123]. MNPs have also been successfully used to deliver therapeutics and reporter genes via a high-field, high-gradient magnetic field to suppress specific genes [122].

MNPs are associated with nucleic acid complexes, added to cell-growing media and subsequently onto the cell surface by applying a magnetic force [124]. Developed by Stephanie Huth and coworkers [125] magnetofection was first used for MNP–naked DNA complexes or MNP–viral vector complexes that were attracted to the bottom by a magnet, placed close below the bottom of a dish. Subsequently, nucleic acid-bound MNPs could be introduced into the cells after their exposition to AMF [126].

Several crucial issues must be addressed when using polymeric MNPs to deliver nucleic acids to cancer cells. First, an AMF should be applied to MNP-DNA conjugates in cell culture to improve the rate of transfection. Second, for in vivo studies, a magnetic field should be applied for the targeted delivery of therapeutic genes. Hence, an AMF can be applied intravenously to target polymeric MNP-DNA complexes in the bloodstream. Once the NPs reach the target site, the genes are released from the particles by either enzymatic cleavage of the crosslinking polymers, pH-dependent reactions, or relapse of the polymer matrix [122]. However, even a slight increase in temperature can result in the overexpression of HSPs both in vitro and in vivo [127]. Therefore, various vectors for gene therapy with the application of an AMF have been constructed using expression systems under the control of an HSP promoter [127]. For example, Tang et al. [128] developed a polymeric Mn-Zn ferrite NP and vector-encoding *β-galactosidase* (*β-gal*) driven by a *HSP70* promoter that highly expressed *β-gal* in tumor cells upon activation with an AMF. Even though *β-gal* expression did not induce anti-oncogenic activities, the results confirmed the potential of using inducible promoters together with MHT [129]. In another study, Yin et al. [130] reported that the binding of HSPs to the promoter acted as a switch to induce the expression of *TNF-α* in human lung adenocarcinoma A549 cells transfected with a *TNF-α*-specific construct with magnetite cationic liposomes and exposed to an AMF (30.6 kA/m, 118 kHz), while the temperature was controlled at 45 °C. Subsequently, TNF-α production was confirmed in cells transfected with the plasmid and exposed to an AMF, indicating that the *HSP70B* promoter was activated by increasing the temperature. Furthermore, the overproduction of TNF-α significantly decreased the viability of transfected A549 cells exposed to an AMF. In the same study, exposure to an AMF for 30 min significantly diminished the tumor volume of transgenic mice [130]. Interestingly, the combination of gene silencing with small interfering RNA (siRNA) and MHT has been used as a diagnostic tool [131,132]. Polymeric magnetic iron oxide NPs conjugated with low molecular weight protamine were used to deliver siRNA directly into tumor cells. Upon application of an external magnetic field, the siRNA concentration was decreased by almost seven-fold but achieved the same diagnostic effect [133].

### 6.4. Photothermal Therapy

The heat of some MNPs is relatively lower due to lower concentrations in the tumor tissue, thus increasing the internal temperature via photothermal therapy was explored as an alternative source [134]. There are two arguments supporting the application of MHT with photothermal therapy. First, although MNP-MHT can be achieved by direct intertumoral injection, particle distribution is significantly limited. Second, the addition of photothermal therapy can reduce the required dosage of irradiation. A synergistic effect was reportedly achieved by combining polymeric MNPs and photothermal nanomaterials with an AMF and laser irradiation [135,136]. For instance, Zhang et al. [137] developed iron-gallic acid (GA) network-based NPs for MRI-guided chemo-photothermal synergistic therapy of tumors. The spatial location and size of the tumor were accurately determined by T1-weighted MRI based on a Fe-GA network. The results of an in vitro study revealed that the Fe-GA network-based NPs were actively endocytosed and induced apoptosis in T1-grade and HepG2 cancer cells. Moreover, the NPs exhibited significant tumor-targeting capability and achieved the highest accumulation in tumor cells after 24 h of incubation [137]. Meanwhile, Ma et al. [138] designed Fe_3_O_4_–Pd Janus NPs (JNPs) with dual-mode MRI/photoacoustic imaging properties for simultaneous magnetic-photo HT and chemodynamic therapy, which achieved relatively higher temperatures through the synchronized application of AMF and laser irradiation. Impressive free radical generation was observed due to the increased magnetic–photo heating of Fe_3_O_4_–Pd JNPs. In addition, ROS production was correlated with the antitumor effects of the Fe_3_O_4_–Pd JNPs in a mouse model of 4T1 orthotopic breast cancer. The Fe_3_O_4_–Pd JNPs with the application of an AMF plus laser irradiation completely inhibited tumor growth without significant adverse effects [138].

### 6.5. Immunotherapy

MNPs have also been applied to enhance the accumulation of active agents to not only kill tumor cells but also sensitize the immune system [5]. The efficacy of MNP-HT combined with immune checkpoint inhibitors has been intensively investigated [26,139]. Interactions with the immune system are influenced by the shape, surface charge, outer shell composition, and aggregation capacity of MHPs. Since the immune system detects extracellular components, partly based on size, NPs are designed to avoid the host immune response. Relatively large carriers are captured by APCs, while smaller particles (<200 nm) can circulate freely in the venous and lymphatic systems for much longer [140]. MNPs trigger the release of various cytokines, both locally and systemically, and recruit immune cells, including neutrophils, DCs, macrophages, natural killer cells, and B and T cells. DCs, for example, present antigens to T cells, which initiate the immune response by activation of CD4+ and CD8+ T cells. Some MNPs can alter the cytotoxic profiles of certain immune cells. Iron oxide NPs shift the polarity of macrophages from the anti-inflammatory M2 phenotype, which promotes uncontrolled tumor growth, to the pro-inflammatory M1 phenotype, which limits tumor growth [141,142]. Notably, most tumor-associated macrophages are the M2 phenotype, which promotes tumor invasion [143]. Regarding the influence of elevated temperature on the tumor environment, HT can induce both apoptotic and necrotic cell death [5]. Necrotic cells release danger-associated molecular patterns (DAMPs) and inflammatory cytokines into the tumor niche. DCs take up tumor antigens and tumor antigen–DAMP complexes for presentation to T cells, which results in the expansion of T cells and the activation of the adaptive anti-tumor immune response via macrophages and natural killer cells [144]. Production of DAMPs can be triggered by dextran-coated SPIONs and the MRI contrast agent Feridex^TM^ (manufactured in Berlex Laboratories (Hanover, NJ, USA)). Specifically, dextran-coated SPIONs activate complement through both the lectin pathway and immunoglobulin M-dependent pathway [145]. Accumulation of SPIONs in U937 monocytes was reported to induce stimulation of the pro-inflammatory Th1 immune response [146]. These discoveries led to the discontinuation of Feridex^TM^ and also highlighted the influence of surface-modified SPIONs on the immune system. For instance, PEG-coated iron oxide NPs induced the expression of interleukin 1β and TNF-α, and caused free radical-dependent toxicity, but not polarization of THP-1 macrophages [147]. On the other hand, polyethylenimine-coated SPIONs were reported to induce M1 polarization of macrophages via Toll-like receptor (TLR)4-mediated signaling pathways and ROS production [148].

The immunostimulatory effects of MNPs have been found to be beneficial against specific types of cancers, such as melanoma [149]. Duval et al. [150] reported that MHT of B16 melanoma cells triggered the overexpression of various immunogenic genes, such as *Hsp-70* and *CXCR3*, as well as the innate immune activators TLR3 and TLR4. Interestingly, MHT combined with other immunotherapeutic approaches might have a synergistic therapeutic effect against oral melanoma [151,152,153]. Immunotherapy combined with HT increased immune cell infiltration and reduced the growth rate of melanoma [154]. The synergistic effect of immunoadjuvants and SPIONs provides a good example of the bimodal activity of immunotherapy and MHT [154]. As an immunoadjuvant, cytosin–-phosphate–guanine (CpG) oligodeoxynucleotides (ODNs) have been implemented in immunotherapy to activate TLR9 signaling pathways to increase the secretion of cytokines that promote the maturation of APCs and improve the Th1 immune response [155]. However, the clinical application of CpG ODNs is limited by nuclease susceptibility, unsatisfactory biodistribution, and systemic adverse effects [156]. Thus, Guo et al. [154] designed magnetic-responsive immunostimulatory NPs (MINPs) with FDA-approved drugs for MRI and anti-cancer immunotherapy. The MINPs were prepared with superparamagnetic iron oxide NPs and CpG encapsulated in monomethoxy PEG-PLGA-poly-L-lysine triblock copolymers. The authors demonstrated that, under an external magnetic field, the MINPs exhibited a magnetic-targeting ability, leading to a high accumulation of SPIONs and CpG ODNs in the tumors. After activation, the MINPs triggered strong anti-tumor immune responses, including DC maturation, cytokine secretion, and tumor infiltration by CD8+ T cells. An immune response was generated by the elimination of primary cancer cells and the inhibition of metastasis [154]. This strategy may potentially be relevant for precise diagnosis and individualized therapy for various tumors.

## 7. Conclusions and Future Perspectives

The main focus of this review was to summarize the applications of NP-mediated MHT for cancer therapy. A brief description of the current strategies used to improve the physicochemical properties and biological activities of MNPs combined with MHT was presented. Various factors, such as the size and shape of polymeric MNPs, viscosity of the medium, and parameters of the applied magnetic field, are crucial for the heating efficiency of MNPs. Furthermore, applications of polymeric NPs for MHT were described. Potential molecular disturbances at the cellular level in response to local increases in the temperature of tumor tissues with the use of MNPs were addressed. In addition, the synergistic effects of MNP-MHT combined with CT, RT, photothermal therapy, gene therapy and immunotherapy were investigated to improve the effectiveness of anticancer treatments. Recent in vitro and in vivo studies demonstrated the high capacity of MNPs for local drug delivery to reduce the viability of cancer cells and inhibit tumor growth. Moreover, the results of cellular and animal studies of MNP-MHT therapy were introduced into clinical trials for the treatment of prostate cancer (clinical trial NCT02033447). Magnetic thermoablation activated by an AMF was shown to effectively destroy cancer cells, while limiting the damage to surrounding healthy tissues. A study completed in January 2015 on 12 male patients with bladder or prostate cancer treated with radical cystoprostatectomy was conducted to assess the location of MNPs because their movement to other areas could have very serious consequences if heat is applied to sensitive, non-cancerous structures around the prostate (back-passage, bladder, sphincter muscle controlling urine flow, and nerves controlling erections). In addition, the fabrication process of MNPs was reviewed to assess the cost effectiveness.

Overall, multi-modal MNPs for drug delivery combined with RT and imaging improved the efficacy of anticancer therapy. Nonetheless, further studies are needed to investigate the biological mechanisms underlying nanoscale heating from the single cell to the whole-body level.

## Figures and Tables

**Figure 1 cancers-16-01156-f001:**
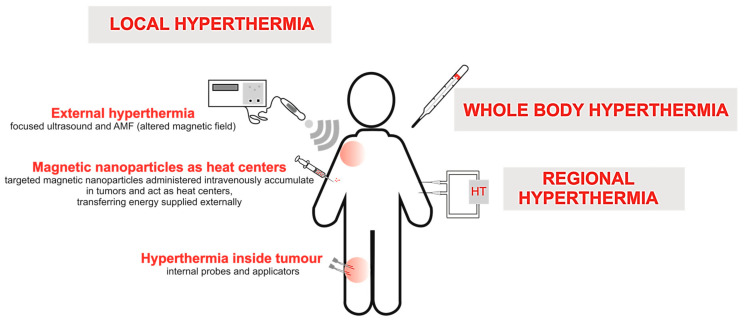
Types of HT used in cancer therapy. Abbreviations: alternating magnetic field (AMF), hyperthermia (HT).

**Figure 2 cancers-16-01156-f002:**
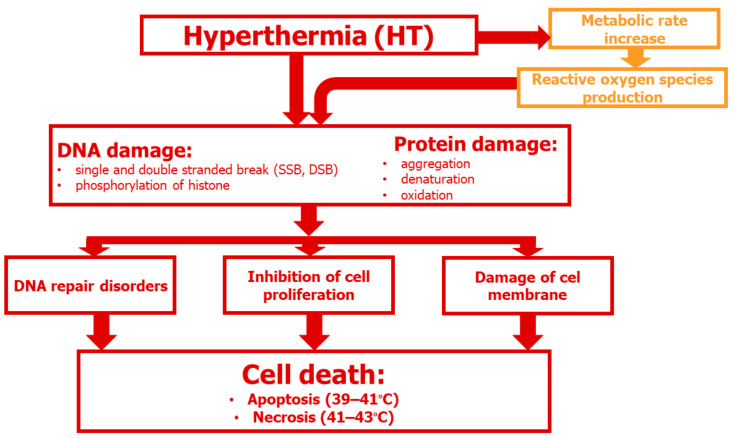
Multidimensional effects of HT at the cellular level.

**Figure 3 cancers-16-01156-f003:**
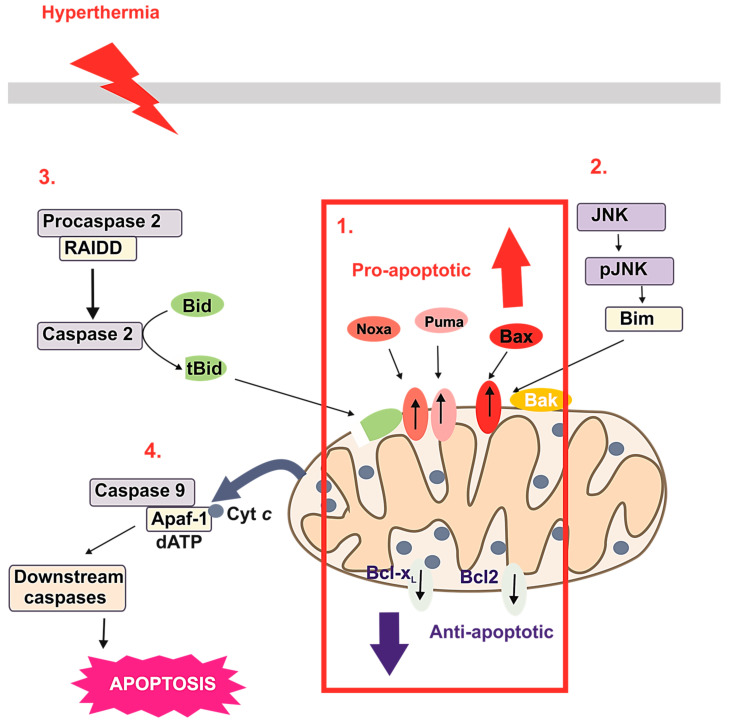
HT disrupts the balance of pro- and anti-apoptosis Bcl-2 family proteins by (**1**) inducing translocation of the pro-apoptotic proteins Bax, Puma, and Noxa to the mitochondria and decreasing levels of the anti-apoptotic proteins Bcl-2 and Bcl-xL. (**2**) HT activates JNKs, which phosphorylate Bim to enhance pro-apoptotic activity. (**3**) Caspase 2 forms a complex with the specific adaptor protein (RAIDD), which activates caspase 2 and cleaves BH3 interacting-domain death agonist (Bid) to tBid. (**4**) Processes (**1**–**3**) trigger cytochrome c release and formation of an apoptosome complex composed of Apaf1 and caspase 9, which activates caspase 9 and initiates a cascade of events leading to apoptosis. Abbreviations: phorbol-12-myristate-13-acetate-induced protein 1 (Noxa), p53 upregulated modulator of apoptosis (Puma), bcl-2-like protein 4 (Bax), B-cell lymphoma-extra-large protein (Bcl -x), B-cell lymphoma 2 (Bcl2), c-Jun N-terminal kinases (JNK), phospho c-Jun N-terminal kinases (p-JNK), Bcl-2-interacting mediator of cell death (Bim), RIP-Associated Protein with A Death Domain (RAIDD), BH3 interacting-domain death agonist (Bid), C-terminal BH3 interacting-domain death agonist (t-Bid), apoptotic protease activating factor 1 (Apaf-1), cytochrome c (cyt c), deoxyadenosine triphosphate (dATP).

**Figure 4 cancers-16-01156-f004:**
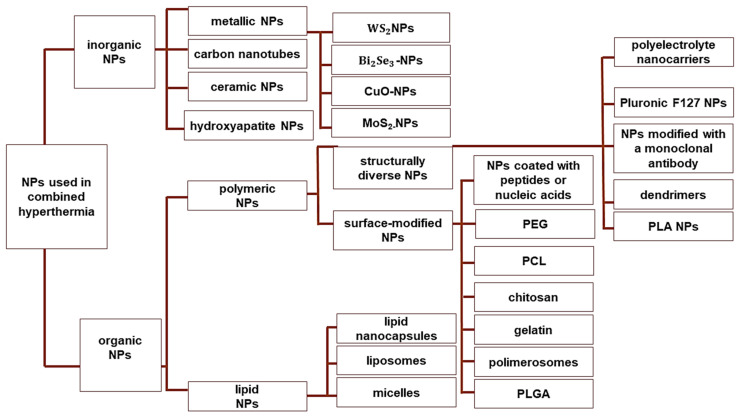
Classification of nanosystems used for HT. Abbreviations: nanoparticles (NPs), tungsten disulfide quantum dots (WS_2_NPs), bismuth selenide nanoparticles (Bi_2_Se_3_-NPs), cupric oxide nanoparticles (CuO-NPs), molybdenum disulfide nanoparticles (MoS_2_-NPs), polyethylene glycol (PEG), poly (caprolactone) (PCL), poly(lactic-co-glycolic acid) (PLGA), Poly(D,L-lactic acid) (PLA).

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
