# Peer review of "Application of Nanoparticles for Magnetic Hyperthermia for Cancer Treatment—The Current State of Knowledge"

_cancers, 2024, doi:10.3390/cancers16061156_

Round 1
Reviewer 1 Report
Comments and Suggestions for Authors
1. The focus of this article is on magnetic particle(MNP)-mediated hyperthermia, but "Figure 1 nanosystems used for HT" shows no MNP.
2. There is a reaction scheme from singlet oxygen to oxygen in Figure 5. What message does the author want to convey?
3. Please add in the principle of alternating magnetic field-induced heat generation by MNPs.
4. Lines 305-307 appears abruptly in the section "Biocompatibility of NP-based drug delivery systems during HT".
5. The references cited are not up-to-date.
6. Lines 512-515 are not described clearly.
Comments on the Quality of English Language
The overall Engllish is fluent, but there are a few places that need improvement in spelling and formating. e.g. "some cell types are more tolerant to heat stress then others ", "in vivo", "in vitro".
Author Response
Dear Reviever#1,
Thank you for sending the revision of our article. We have now revised the manuscript according to your remarks as detailed below in italics. We are appreciate for your comments which have resulted in an improved manuscript. We have submitted the revised version showing all changes in the file labelled RED, whereas the clean updated version is found in the file labelled BLACK. We hope that you find the revised manuscript acceptable for publication in Cancers in its current form.
Best regards,
Marzena Szwed
Corresponding author
Marzena Szwed (marzena.szwed@biol.uni.lodz.pl)
Department of Medical Biophysics
Faculty of Biology and Environmental Protection
141/143 Pomorska St.
90-236 Lodz
Poland

Reviewer 2 Report
Comments and Suggestions for Authors
The submitted manuscript describes a summary of recent advancements in a hyperthermia (HT) with the use of magnetic nanoparticles to locally increase temperature and promote cell death. In addition, the submitted manuscript summarizes recent developments of nanocarriers as nanoparticles-based drug delivery systems. Although the main text of the submitted manuscript has been written well, illustrations of several Figures should be redrawn. The reviewer thinks that the submitted manuscript would be suitable for publication for “cancers” after the authors modify several corrections.
Information shown in Figure 1 is not sufficient. Since the authors mention “three main types of HT have been applied in clinical practice (Fig. 1): whole-body, regional, and local.” (lane 65 to 66), Figure 1 should show three main types of HT. The use of color in Figure 2 is not good. It is hard to see the illustration of Figure 2 because of the use of vivid colors. In addition, the reviewer was not able to understand what the arrows indicate. The abbreviation of “RAIDD” is not required. (lane 173) The resolution of Figure 3 should be improved. The reviewer was not able to understand what the authors intend in Figure 4. Why are three colors (green, blue, and brown) used in the classification of NPs used for HT? The molecular formula of Fe2O3 (lane 448) and Fe3O4 (lane 556, 560, and 562) should be described accurately. “Subchapter 7.3. Gene therapy” (lane 505) should be plain. The display of reference (lane 537) should be corrected. The space of lane 566 should be removed. “TM” (lane 593) is a trademark. “TM” should be superscript. Information on MNPs (lane 606) is not required. The phrases “in vitro” and “in vivo” should be italic and the word “and” should be plain (lane 637). Is the copyright of Figure 5 OK?
Author Response
Dear Reviever#2, Thank you for sending the revision of our article. We have now revised the manuscript according to your remarks as detailed below in italics. We are appreciate for your comments which have resulted in an improved manuscript. We have submitted the revised version showing all changes in the file labelled RED, whereas the clean updated version is found in the file labelled BLACK. We hope that you find the revised manuscript acceptable for publication in Cancers in its current form.
Best regards,
Marzena Szwed
Corresponding author
Marzena Szwed (marzena.szwed@biol.uni.lodz.pl)
Department of Medical Biophysics
Faculty of Biology and Environmental Protection
141/143 Pomorska St.
90-236 Lodz
Poland
